# Mucosal Antibody Response to SARS-CoV-2 in Paediatric and Adult Patients: A Longitudinal Study

**DOI:** 10.3390/pathogens11040397

**Published:** 2022-03-24

**Authors:** Renee W. Y. Chan, Kate C. C. Chan, Grace C. Y. Lui, Joseph G. S. Tsun, Kathy Y. Y. Chan, Jasmine S. K. Yip, Shaojun Liu, Michelle W. L. Yu, Rita W. Y. Ng, Kelvin K. L. Chong, Maggie H. Wang, Paul K. S. Chan, Albert M. Li, Hugh Simon Lam

**Affiliations:** 1Department of Paediatrics, Faculty of Medicine, The Chinese University of Hong Kong, Hong Kong, China; katechan@cuhk.edu.hk (K.C.C.C.); josephtsun@cuhk.edu.hk (J.G.S.T.); kathyyychan@cuhk.edu.hk (K.Y.Y.C.); jasmineyipsinki@link.cuhk.edu.hk (J.S.K.Y.); 1155152004@link.cuhk.edu.hk (S.L.); ywl592@ha.org.hk (M.W.L.Y.); albertmli@cuhk.edu.hk (A.M.L.); hshslam@cuhk.edu.hk (H.S.L.); 2Laboratory for Paediatric Respiratory Research, Li Ka Shing Institute of Health Sciences, Faculty of Medicine, The Chinese University of Hong Kong, Hong Kong, China; 3CUHK-UMCU Joint Research Laboratory of Respiratory Virus & Immunobiology, Department of Paediatrics, Faculty of Medicine, The Chinese University of Hong Kong, Hong Kong, China; 4Hong Kong Hub of Paediatric Excellence, The Chinese University of Hong Kong, Hong Kong, China; 5Department of Medicine and Therapeutics, Faculty of Medicine, The Chinese University of Hong Kong, Hong Kong, China; gracelui@cuhk.edu.hk; 6Department of Paediatrics, Prince of Wales Hospital, New Territories, Hong Kong, China; 7Department of Microbiology, Faculty of Medicine, The Chinese University of Hong Kong, Hong Kong, China; ritang@cuhk.edu.hk (R.W.Y.N.); paulkschan@cuhk.edu.hk (P.K.S.C.); 8Department of Ophthalmology and Visual Sciences, Faculty of Medicine, The Chinese University of Hong Kong, Hong Kong, China; chongkamlung@cuhk.edu.hk; 9The Jockey Club School of Public Health and Primary Care, Faculty of Medicine, The Chinese University of Hong Kong, Hong Kong, China; maggiew@cuhk.edu.hk

**Keywords:** SARS-CoV-2, mucosal antibody, paediatric, specific IgA, specific IgG

## Abstract

Background: SARS-CoV-2 enters the body through inhalation or self-inoculation to mucosal surfaces. The kinetics of the ocular and nasal mucosal-specific-immunoglobulin A(IgA) responses remain under-studied. Methods: Conjunctival fluid (CF, *n* = 140) and nasal epithelial lining fluid (NELF, *n* = 424) obtained by paper strips and plasma (*n* = 153) were collected longitudinally from SARS-CoV-2 paediatric (*n* = 34) and adult (*n* = 47) patients. The SARS-CoV-2 spike protein 1(S1)-specific mucosal antibody levels in COVID-19 patients, from hospital admission to six months post-diagnosis, were assessed. Results: The mucosal antibody was IgA-predominant. In the NELF of asymptomatic paediatric patients, S1-specific IgA was induced as early as the first four days post-diagnosis. Their plasma S1-specific IgG levels were higher than in symptomatic patients in the second week after diagnosis. The IgA and IgG levels correlated positively with the surrogate neutralization readout. The detectable NELF “receptor-blocking” S1-specific IgA in the first week after diagnosis correlated with a rapid decline in viral load. Conclusions: Early and intense nasal S1-specific IgA levels link to a rapid decrease in viral load. Our results provide insights into the role of mucosal immunity in SARS-CoV-2 exposure and protection. There may be a role of NELF IgA in the screening and diagnosis of SARS-CoV-2 infection.

## 1. Introduction

Severe acute respiratory syndrome coronavirus 2 (SARS-CoV-2) causes coronavirus disease (COVID-19) [1]. SARS-CoV-2 interacts with the angiotensin-converting enzyme 2 (ACE2) expressed by the nasal epithelia to enable the entry and infection of neighboring epithelial cells [1,2], while the conjunctival goblet cell is an alternative portal of entry [3,4]. Therefore, examining the mucosal antibodies of COVID-19 patients may provide a more thorough discernment of viral-host interaction and the underlying immunopathology. Although mucosal immunity plays an important role in SARS-CoV-2 infection, most studies have focused on systemic immunity [5,6,7]. There is a paucity of studies investigating the SARS-CoV-2-specific antibodies of the conjunctival and respiratory mucosae.

Mucosal immunity is achieved by innate and acquired immune responses [8,9]. Viral antigens acquired locally in the conjunctival and nasal epithelia are processed in the conjunctiva-associated lymphoid tissue (CALT) [10] and nasopharyngeal-associated lymphoid tissue (NALT), respectively [11]. Meanwhile, these lymphoid tissues generate IgA-producing mucosal B cells that express homing receptors for efficient trafficking to the mucosal effector site [12,13]. Secretory IgA is a potent dimeric IgA that is found on mucosal surfaces [14], and provides broad protection, due to its high avidity [15]. It is responsible for agglutinating and neutralizing viruses in the respiratory tract, in the respiratory cell, and within the lamina propria beneath the epithelium. The dimeric form is found to be fifteen-fold more potent than its monomeric counterpart in plasma [16]. Moreover, it provides effective immunity against infection when compared with its IgG isotype [17] and in the investigation of antibody neutralization in the natural disease course of SARS-CoV-2 infection [18].

The early and intense induction of serological IgA in COVID-19 patients has been reported previously [6]. Sterlin et al. documented that the first wave of circulating IgA-expressing plasmablasts precedes the IgG-expressing cells [18]. Cervia et al. reported that mild disease or low antigen exposure might stimulate a mucosal SARS-CoV-2-specific IgA response, which could be accompanied by the absence, presence, or delay of systemic virus-specific IgA production [5]. Such a pattern appears to be particularly prevalent in younger individuals and might explain why children commonly present with asymptomatic or mild SARS-CoV-2 infection. However, this hypothesis requires further supportive evidence from longitudinal studies in both children and adults. Our study aimed to evaluate longitudinal mucosal SARS-CoV-2-specific antibody levels and their neutralizing effect, to address this knowledge gap.

In our study, conjunctival fluid (CF) samples were collected using a technique similar to Schirmer’s test [19,20], while the nasal epithelial lining fluid (NELF) samples were collected with nasal strips [21,22]. These methods are standardized and exhibit extended sample stability, even when stored at room temperature [23], thus overcoming the problem with sample validity that is associated with current mucosal sampling methods, namely, nasal swabs or irrigation.

## 2. Results

### 2.1. Demographics of the Subjects

Thirty-four paediatric patients and forty-seven adult patients participated in this study. All subjects tested negative to other respiratory pathogens in the multiplex panel at the point of admission. The median age was 12.5 years old (ranging from 6 to 17 years of age) for the paediatric group and 61 years old (ranging from 18 to 88 years of age) for the adult group, with 32% and 38% of male subjects in the respective groups (Figure 1B). Except for one patient who had moderate disease (with pneumonia), 19 of the paediatric subjects exhibited mild disease (without pneumonia), while 14 of them were asymptomatic. Moreover, as we just had one paediatric patient who had moderate disease severity, the data generated from this subject were excluded from the statistical analysis. As there were only two critically ill adult patients (requiring mechanical ventilation), we pooled the severe and critically ill groups together for the final analysis. A total of 140 CF, 424 NELF, and 151 plasma samples were collected (Figure 1B).

### 2.2. SARS-CoV-2 S1-Specific IgA Dominated Conjunctival and Nasal Epithelial Lining Fluids

S1-specific IgA (referred to as “IgA” hereafter) were detected in 50% of the CF, in 54% of the NELF and in 43% of the plasma samples of the paediatric patients within the first four days of disease diagnosis (Appendix A and Appendix A). S1-specific IgG (referred to as “IgG” hereafter) was not detected in the CF at any of the time points (Figure 2B). A minority of the NELF samples tested positive for IgG twelve days after disease diagnosis (Figure 2F,H), while the plasma IgA was detected earlier than the IgG. In contrast, IgG became dominant by three months after diagnosis (Appendix A). At six months post-diagnosis, 59% of the CF and 50% of the NELF remained IgA-positive, while 71% and 86% of the plasma samples remained IgA- and IgG-positive, respectively.

In the NELF of adult patients, IgA was also the dominant isotype (Appendix A). Overall, 21% and 31% of adult NELF collected on days 0 to 4 and days 5 to 9 post-diagnosis were IgA-positive. At six months post-diagnosis, 58% of the NELF remained IgA-positive. However, we did not have paired plasma samples to assess the longevity of plasma IgA and IgG (Appendix A).

### 2.3. Symptomatic COVID-19 Paediatric Patients Had a Higher Level of IgA in CF

Symptomatic children had higher levels of IgA at 12 to 16, 26 to 30, and 169 to 197 days post-diagnosis (*p* = 0.0365, 0.0252 and 0.0431), respectively (Appendix A, Figure 2A) than asymptomatic children, while no IgG was detected (Figure 2A). In total, 93% of the paediatric patients who provided their CF samples for longitudinal measurements were IgA-positive for at least one time point.

### 2.4. Asymptomatic Paediatric Patients Had an Early Induction of IgA in Their Nasal Mucosa and a Higher Level of IgG in Their Plasma

The asymptomatic paediatric patients’ NELF contained significantly higher levels of IgA (median S/C ratio = 2.9 vs. 0.7, *p* = 0.0017, Figure 2D) at 0 to 4 days post-diagnosis than their symptomatic counterparts. Moreover, a significantly higher level of IgG was detected in the plasma of the asymptomatic than the symptomatic paediatric subjects at 12 to 16 days post-diagnosis (median S/C ratio = 7.1 vs. 1.3, *p* = 0.0364, Figure 2F). Interestingly, a higher percentage of the symptomatic subjects retained IgA in their plasma after 3 months (asymptomatic = 25% vs. symptomatic = 80%, *p* = 0.0949) and after 6 months (asymptomatic = 25% vs. symptomatic = 90%, *p* = 0.0410) after diagnosis (Appendix A, Figure 2K).

### 2.5. A Similar Percentage of IgA-Positive NELF in Adult Patients of Different Disease Severity

In adults, a robust induction of IgA was detected in the NELF after 9 days post-diagnosis (Figure 3A), while there were no statistical differences in their IgA levels and the percentage of NELF IgA-positive patients among the three disease severity groups (Figure 3A,C and Appendix A). Moreover, although differences in the levels of NELF IgG levels between adult patients with mild disease, compared to those with moderate disease (*p* = 0.0249), and severe and critically ill patients (*p* = 0.0461) at 5 to 9 days post-diagnosis were significant, the median of these samples was below the detection limit (Figure 3B,D and Appendix A).

### 2.6. Early Induction of IgA in the Plasma of Adult Patients with Mild Disease

With the limited number of plasma samples collected, we focused our analysis on the first two weeks post-diagnosis. There were significantly higher levels of plasma IgA (Figure 3C and Appendix A, median S/C ratio = 2.2 vs 0.2, *p* = 0.0476) in patients with mild disease than those with a severe or critical illness at 5 to 9 days post-diagnosis. In contrast, IgG was not detectable at 0 to 4 days post-diagnosis and only became detectable by 12 to 16 days post-diagnosis (Figure 3E and Appendix A).

### 2.7. Symptomatic Paediatric Patients Had a Higher Level of IgA in Their Plasma during the Early Phase of SARS-CoV-2 Infection Than Adult Patients with Mild Disease

We compared the mucosal, serological antibody responses, and viral load between symptomatic paediatric (*n* = 19) and adult patients with mild disease (*n* = 18). Though there was a trend toward higher percentages of IgA-positive NELF samples in symptomatic paediatric patients at 5 to 9 (63% vs. 47%) and 12 to 16 days post-diagnosis (95% vs. 89%) than adult patients with mild disease (Table 1), the two groups had similar levels of NELF IgA.

Nevertheless, symptomatic paediatric patients had a higher plasma level of IgA (median S/C ratio = 1.1 vs 0.4, *p* = 0.0082) and a trend toward higher percentages of IgA-positive plasma samples on 0 to 4 (50% vs. 14%), 5 to 9 (100% vs. 63%) and 12 to 16 days post-diagnosis (100% vs. 75%) than adult patients with mild disease (Table 2). No differences in the percentage of IgG-positive plasma samples were found between the two age groups.

The antibody levels in the S/C ratios at the same time point were compared using the Mann–Whitney test, while the percentages of positive samples were compared by Fisher’s exact test. *p* values smaller than 0.05 are bolded, *p* values > 0.9999 are represented by ns (not significant), while dashes mean that there are no data for comparisons.

### 2.8. Asymptomatic Paediatric Patients Had a Lower Viral Load during Admission, while Adult Patients with Mild Disease Had a Sharp Reduction in Viral Load in the First Week after Diagnosis

Asymptomatic paediatric subjects had a significantly lower viral load than their symptomatic counterparts at 0 to 4 (median CT = 34.0 vs. 20.0, *p* < 0.0001) and 5 to 9 days post-diagnosis (median CT = 40.0 vs. 32.3, *p* = 0.0234) (Table 3 and Figure 4A) while in adults, similar viral loads were detected among the three severity groups at admission (Table 3 and Figure 4B). Moreover, asymptomatic paediatric patients had a significantly lower viral load than all adult patients in the first 9 days post-diagnosis (Table 3, *p* < 0.0001). Symptomatic paediatric patients also had a lower viral load than adult patients with moderate diseases (median CT = 32.3 vs. 27.8, *p* = 0.0466) and those who were severe and critically ill (median CT = 32.3 vs. 24.8, *p* = 0.0007) by 5 to 9 days post-diagnosis and those who were severe and critically ill (median CT = 35.0 vs. 28.9, *p* = 0.0361) by 12 to 16 days post-diagnosis.

Viral loads in asymptomatic paediatric patients decreased from days 0 to 4 (median CT = 34.0) and days 5 to 9 (median CT = 40) post-diagnosis (Table 4, *p* = 0.0564). In comparison, the viral loads of the symptomatic paediatric patients did not change significantly within the first nine days of post-diagnosis and persisted until 12 to 16 days post-diagnosis (median CT = 35). In adults, the viral loads of patients in the mild and moderate groups declined over the first three weeks, while the viral loads in the severe and critically ill patients remained robust from 0 to 4 (median CT = 21.0) to 19-to-23 (median CT = 29.6) days post-diagnosis (Table 4). As patients with mild disease were discharged earlier, no CT values were available at the subsequent time points for comparison.

### 2.9. IgA Levels Correlated Positively with the SARS-CoV-2-Neutralizing Effect of the Mucosal and Plasma Samples

Fifty-five percent (18/33) of the CF samples showed a neutralizing effect. All these eighteen CF samples were IgA-positive. However, IgA positivity did not translate directly to a neutralizing effect; 39% (12/31) of IgA-positive CF did not inhibit the binding of ACE-2 with the SARS-CoV-2 receptor-binding domain. Nevertheless, higher levels of IgA were detected in the receptor-blocking samples (median S/C ratio = 7.926) than the non-receptor-blocking CF samples (median S/C ratio = 2.648, *p* = 0.0043, Figure 5E). A significant positive correlation was found between the IgA levels in CF (r = 0.6094, *p* = 0.0005, Figure 5A) and NAb levels.

Sixty-eight percent (53/78) of the NELF samples showed a neutralizing effect; all these 53 NELF samples were also IgA-positive. In comparison, only 25% (13/53) of the NAb-positive samples were also IgG-positive. Higher levels of IgA were detected in the receptor-blocking samples (median S/C ratio = 10.250) than the non-receptor-blocking NELF samples (median S/C ratio = 4.140, *p* = 0.0009, Figure 5E). Significant positive correlations were found between the IgA (r = 0.5498, *p* < 0.0001, Figure 5B) and IgG levels (r = 0.6170, *p* < 0.0001, Figure 5B) and the NAb level in the NELF samples.

Sixty-eight percent (53/77) of the plasma samples were SARS-CoV-2 receptor-blocking. Ninety-eight percent (52/53) and 89% (47/53) of the receptor-blocking plasma samples were IgA- and IgG-positive, respectively. In contrast to NELF, it is worth noting that a high percentage of receptor-blocking plasma samples, 87% (46/53), were positive for both IgA and IgG. Significant positive correlations were found between the IgA (r = 0.8585; *p* < 0.0001) and IgG levels (r = 0.9497; *p* < 0.0001) with the NAb level in the plasma samples (Figure 5C).

### 2.10. Receptor-Blocking IgA in NELF, Detected in the First Week of Diagnosis, Correlated with a Rapid Decrease in Viral Load

ROC analysis was used to determine a threshold for predicting the neutralizing effect on the first-week NELF samples (Figure 6A). When the IgA S/C level was above the threshold, the neutralizing effect of the NELF sample was predicted and vice versa. Using a fixed-effect regression model, we showed that paediatric patients who had NELF IgA levels above the threshold level of the S/C ratio at 4.386 in the first week after diagnosis had a more rapid decline in viral load than those who did not (*p* = 0.002, Figure 6B).

## 3. Discussion

SARS-CoV-2 can enter the body via inhalation or by self-inoculation directly onto the mucosal surfaces. While it is known that mucosal IgA plays a vital role in the first line of defense against pathogens, including SARS-CoV-2 [24], the kinetics of the ocular and nasal mucosal-specific IgA responses remain under-studied. This study recruited paediatric and adult COVID-19 patients and profiled their S1-specific mucosal antibody levels longitudinally, from hospital admission to six months post-diagnosis. The timing of the SARS-CoV-2 antibodies’ production, immunoglobulin isotypes, concentrations, receptor-blocking potency, antibody longevity, and relevance in the different age groups were studied.

Our longitudinal profiling of the antibodies reveals the IgA-dominant mucosal response in COVID-19 patients. Though none of our paediatric patients had clinical symptoms or signs suggestive of conjunctivitis, in our CF study, there was a surge of IgA from 0 to 4 days to 12 to 16 days post-diagnosis in the symptomatic patients and significantly higher IgA levels in symptomatic than in asymptomatic patients during the 2nd and 4th weeks post-diagnosis, while no IgG was detectable in the CF. Our result infers a more intense involvement of CALT in symptomatic paediatric patients. This finding is possibly due to an anterior chamber-associated immune deviation of the eye that tends to eliminate B cells which produce complement-fixing antibodies, e.g., IgG [25]. We focused on the CF detection in paediatric patients, as the development of CALT starts in childhood and increases till the age of 10 years, then subsequently declines with age [26]. A similar measurement was conducted in twenty-eight COVID-19 adult patients (mean age = 66, age range 18–89) in which only 35.7% of them evidenced the presence of ocular anti-SARS-CoV-2 IgA [27]. In parallel with this adult study, the ocular IgA persists in our paediatric subjects. Seventy percent of the CF from symptomatic patients remained IgA-positive by six months post-diagnosis, compared with to 43% in the asymptomatic children. Whether this could result in better protection from re-infection through eye mucosae requires further investigations.

The IgA response is compartmentalized, whereas an opposite pattern was observed in the nasal mucosa. In the first four days of diagnosis, a more prominent IgA was detected in the asymptomatic paediatric patients. While the NELF IgA level correlated with the NELF receptor-blocking potency in the surrogate neutralization assay, the biological correlation of the NELF IgA was further investigated. Upon the detection of “receptor-blocking” NELF IgA in the first week after diagnosis, a more rapid decline of viral load was observed. This pattern supports the hypothesis that the early response of SARS-CoV-2-specific-IgA limits the replication of SARS-CoV-2, leading to a mild to asymptomatic disease course. It also infers that a vaccine pertaining to inducing nasal immune responses and memory cells could have additional advantages over traditional ones that only induce circulating antibody responses [28].

Our results suggest that the mucosal IgA response is localized and demonstrated that IgA, when produced systemically, is not transported into the secretions as previously described [29]. An early and robust NELF IgA was induced in asymptomatic paediatric patients rather than in paediatric patients with mild disease. In addition, all adult patients had low NELF IgA levels on 0 to 4 days post-diagnosis, while the first statistically significant increase of IgA in adult patients was detected only by 12 to 16 days post-diagnosis. The early and sufficient response of NELF IgA infers protective effects, such as a shorter viral-shedding period and a milder disease presentation. This is also reported in human 229E infection [30]. In contrast, the CF and blood IgA levels detected on or beyond 12 days post-diagnosis were associated with disease severity, together with the extended response of the conjunctival and plasma antibodies, as reported in other studies [31,32,32].

From the diagnostic point of view, as the mucosal antibody is detectable before symptoms occur and, in a virus-exposed individual who appears negative in a serological test [33], the mucosal antibody test is a promising screening tool for asymptomatic individuals, especially during the early stage of the disease. Indeed, NELF IgA provides a sensitive readout as evidence of SARS-CoV-2 infection in the first month of disease diagnosis. All subjects were positive for NELF IgA for at least one-time point. However, the duration from a prior SARS-CoV-2 infection inevitably determines the sensitivity of both mucosal and serological antibody levels [32], among which mucosal IgA is less affected [18]. The NELF IgA was still detectable in at least 50% of the COVID-19 patients after three months (day 83-to-99) of diagnosis. Importantly, the non-invasive nature and the validity of the paper strips used in our study would allow repeated self-tests without blood sampling [23,34].

Lastly, as this study focused on the mucosal responses and the number of plasma samples collected in paediatric subjects, it depended on the need for clinical management and monitoring, rather than being research-driven. Therefore, there was a higher number of plasma samples in the symptomatic group of paediatric patients. The comparison made here would be less comprehensive when compared to the existing literature that focused on the serological measurements. The evidence from serological studies suggests that the induction of SARS-CoV-2-specific antibodies is positively associated with the disease severity [6,7,35]. A study comparing the SARS-CoV-2-specific IgM, IgG, and neutralizing antibody in the sera of asymptomatic patients with sex-, age- and comorbidity-matched, mildly symptomatic patients during the acute and early convalescent phases identified that significantly lower IgG levels were detected in asymptomatic adult patients [36]. In our study with a smaller sample size, we found subtly higher levels of IgA in the plasma of adult patients with mild disease than in those who were severely and critically ill at 5 to 9 days post-diagnosis. However, no noticeable change in the plasma IgG was observed among the three adult severity groups. Apart from the quantity of the S1-specific antibody, Hachim et al. found that infected children had lower spike and nucleocapsid antibody levels than adults. Interestingly, infected paediatric subjects had a more expanded response to the accessory proteins [37].

One of the major limitations of this study was that we determined the SARS-CoV-2-specific antibody only in terms of its IgA and IgG isotypes and only in those against S1. However, these assays only provided a semi-quantitative readout that could be normalized by the absolute total IgA and IgG concentrations. We drew the conclusion from the assumption that the total IgA between paediatric and adult COVID-19 patients could be comparable from the fact that the IgA level in secretions matured early and reached adult levels at 6–12 months [29].

Moreover, the diversity of the antibody responses to other SARS-CoV-2 viral antigens was not thoroughly evaluated. These non-receptor-blocking antibodies are detectable in COVID-19 patients’ plasma early in the disease course and counteract the viral inhibition in host antiviral effects [37]. It would be of paramount importance to characterize the antibody diversities in the mucosal fluids and to determine their immunological relevance in divergent disease outcomes.

We are aware of the recent reports about the similarity of the S2 portion of OC43 to SARS-CoV-2. Therefore, the presence of or the re-induction of the OC43 antibody in our patient groups (especially in the paediatric group), would be a possible reason for the attenuated severity of SARS-CoV-2 infection, as discussed in the recent literature [38,39]. However, human coronaviruses (OC43, 229E, NL63, and HKU1) were not the listed agents in routine detection before the SARS-CoV-2 era, and human coronavirus infection is generally mild and might not lead to hospitalization. Therefore, we do not have further information about their OC43 infection history.

Another limitation of our study was the lack of a cell-based plaque reduction assay, which required biosafety level-3 facilities. In addition, the direct measurement of the S1-specific antibody might not confer immunity, while the surrogate neutralization assay is expensive when a huge number of samples is to be tested. Here, we derived a threshold of IgA levels that is predicted to have “receptor-blocking” potency. We applied this threshold to the rest of the NELF IgA measurements and converted it to a biologically relevant observation. Patients with “receptor-blocking” NELF within the first 7 days of diagnosis had a more rapid decline in viral load. This threshold translates to a workable alternative for laboratories that lack the necessary facilities and diagnostic utilities. Finally, the small number of severe and critically ill patients limits the generalizability of our results to severely affected patients.

## 4. Materials and Methods

### 4.1. Subject Recruitment

The presence of SARS-CoV-2 infection was confirmed by the detection of the E gene using the LightMix^®^ SarbecoV Kit (Tib-Molbiol, 40-0776-10) following the manufacturer’s instructions. Paediatric and adult patients who were hospitalized with COVID-19 were recruited prospectively if they were within four days of their first RT-PCR-positive result, from June 2020 to January 2021. All patients were unvaccinated and without known prior SARS-CoV-2 infection. Day 0 was considered as the first day of being SARS-CoV-2-positive for asymptomatic patients and the first day of symptoms for symptomatic patients, respectively.

The duration after infection plays a major role in the immune response in the case of SARS-CoV-2. We provided the median and interquartile range of the period between the onset of symptoms and the day of diagnosis in Appendix A. Patients were ready for discharge when they consecutively tested negative for SARS-CoV-2 by RT-PCR or had a viral threshold cycle (CT) value of above 32 and tested positive for nucleocapsid specific serum IgG with a chemiluminescent microparticle immunoassay assay. Longitudinal biospecimen collections were conducted at seven time points during the in-patient period and after discharge.

The biospecimens (CF, NELF, and plasma) from paediatric patients were collected by healthcare workers during the follow-up appointments, while NELF was collected by adult patients via self-collection (Figure 1A).

### 4.2. Severity Scoring

Disease severity was categorized as described in the World Health Organization’s COVID-19 clinical management living guidance [40]. The severity of the asymptomatic patient was denoted as 0, the disease severity of the symptomatic subjects was categorized into mild (score 1, where the clinical symptoms were light, and there was no sign of pneumonia on imaging), moderate (score 2, with fever, respiratory tract problems and other symptoms, with imaging suggesting pneumonia), severe (score 3, coinciding with any of the following: (1) respiratory distress, respiration rate (RR) ≥ 30 times/min; (2) oxygen saturation of ≤ 93% in the resting state; (3) PaO_2_ / FiO_2_ ≤ 300 mmHg (1 mmHg = 0.133 kPa)) and critically ill (score 4, coinciding with any of the following: (1) respiratory failure occurs and mechanical ventilation is required; (2) shock; (3) the patient develops other organ failure and needs ICU monitoring and treatment).

### 4.3. Asymptomatic Paediatric Subject Recruitment

During the study period, the symptomatic paediatric COVID-19 patients were mainly identified by a compulsory screening scheme in the airport upon their arrival (*n* = 1) OR by the compulsory screening of a household member or close contact with a confirmed case (*n* = 7). Moreover, for those who were identified during the quarantine period in the designated quarantine facility upon arrival from overseas (*n* = 3), or as the household member or close contact of a confirmed case (*n* = 3), their first day of a PCR-positive result would be very close to the day of infection. During the study period, the HKSAR government reacted to every identified COVID-19 case and tested their household members and close contacts within a day. We summarized the median and interquartile range of the period between the onset of symptoms and the day of diagnosis of our symptomatic cases for clearer data interpretation in Appendix A.

### 4.4. CF and NELF Collection

The sampling of CF was conducted on both eyes, with a technique similar to performing the Schirmer’s test [20,28]. The paper strip (I-Dew Tear Strips, Entod Research Cell, United Kingdom) was inserted into the lower conjunctival sac and collected after the fluid reached the 25 mm mark [20]. The nasal strip, made of Leukosorb, was inserted into each nostril after 100 uL of sterile saline was instilled, as described [23,28], followed by a one-minute nose pinch. All strips were collected and transferred to a dry sterile collection tube and eluted within 24 h after collection.

### 4.5. Elution of CF and NELF and the Preparation of Plasma

To elute, ocular or nasal strips were soaked in 300 uL of phosphate-buffered saline (PBS) on ice. The solution and the strips were transferred to a Costar Spin-X (CLS9301) and centrifuged at 4 °C to elute the CF or NELF. Then, 3 mL of blood was collected by venepuncture and transferred into an EDTA blood tube. Plasma samples were separated by centrifugation at 4 °C at 2000 g for 20 min. The specimens were aliquoted into small volume vials for the downstream analysis of SARS-CoV-2-specific Ig panels and neutralization test and stored at −20 °C until analysis.

### 4.6. Measurement of SARS-CoV-2 Spike Protein-Specific IgA and IgG

Semi-quantitative measurements of SARS-CoV-2 spike protein (S1 domain)-specific Ig ELISA Kits (Euroimmun, EI 2606-9601 A and EI 2606-9601 G) were used. For this measurement, 1:10 diluted-CF and NELF, as well as 1:100 diluted plasma, were assayed as per the manufacturer’s instructions and analyzed with a Synergy HTX Multi-Mode Reader. A semi-quantitative readout was used for the ratio between the sample and the calibrator’s optical density (OD). Data were expressed in the sample/calibrator (S/C) ratio, where a value of ≥ 1.1 was considered positive.

### 4.7. Measurement of SARS-CoV-2 Neutralizing Antibody (NAb)

A blocking enzyme-linked immunosorbent assay (GenScript, L00847) was employed as a surrogate of the neutralization test. Briefly, undiluted CF, NELF, 1:9 diluted plasma samples, and controls were processed as per the manufacturer’s instructions. Samples that gave a signal inhibition of ≥ 30% were considered to be SARS-CoV-2 NAb-positive.

### 4.8. Statistical Analysis

The demographic variables of the subjects were compared between disease severity groups using the Mann–Whitney test, Kruskal–Wallis test, and Fisher’s exact test, as appropriate. For the immunoglobulin profiles, differences between genders and age groups were evaluated using the Mann–Whitney test, while differences in disease severity groups and time points were tested using the Kruskal–Wallis test, followed by Dunn’s multiple comparisons test. The correlation of the S/C ratio of the specific immunoglobulins with the percentage of signal inhibition in the surrogate neutralization test was examined using Spearman’s correlation test. Using the ROC curve, a threshold was derived using the Youden Index calculation, with the assumption that sensitivity and specificity hold equal diagnostic importance, while *J* provides an optimal threshold. J=Sensitivity+Specificity−1, such that the threshold used provides the maximum, *J.* The threshold provided the basis for predicting the receptor-blocking effect. When the S1-specific IgA S/C level was above the threshold, the receptor-blocking effect of the NELF sample was predicted, and vice versa. Considering the dependence of repeated measurements of the same subjects, a fixed-effect regression model was used to determine the differences in the viral loads and whether they declined over time between samples from paediatrics patients who had neutralizing antibody at the first week of diagnosis. All statistical tests were performed using GraphPad version 9.1.2 for the macOS SPSS version 25. Differences were considered statistically significant at *p* < 0.05.

## 5. Conclusions

In conclusion, our longitudinal study in COVID-19 paediatric and adult patients depicts an independent landscape of mucosal and systemic antibody responses. The higher mucosal IgA levels are protective and are more commonly found in asymptomatic subjects. The high mucosal IgA level is in great contrast to our current understanding of the antibody profile in the circulation, in which higher antibody levels are linked with severity. The differential intensity of the secretory IgA invites attention, particularly when characterizing the mucosal antibody spectrum and their implications for clinical presentations. Our finding provides an important dimension for the diagnostic use of the mucosal antibody measurements, especially for individuals at an early phase of infection and in asymptomatic patients.

## Figures and Tables

**Figure 1 pathogens-11-00397-f001:**
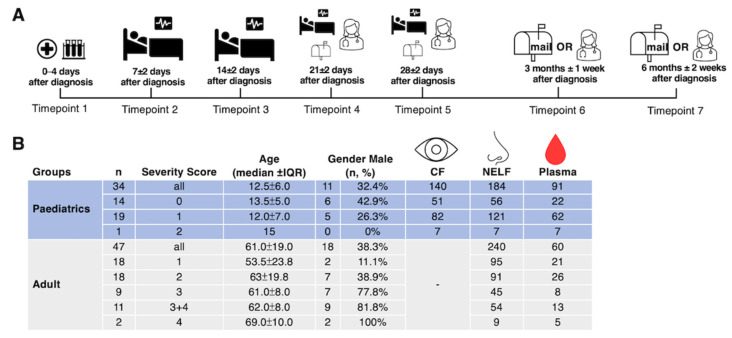
Study design and demographics. (**A**) A longitudinal sample collection, from the day of diagnosis (disease onset or the first day of a SARS-CoV-2 PCR positive result, whichever was earlier) to six months post-diagnosis, was conducted by healthcare workers during hospitalization and follow-up consultations for paediatric patients. Adult patients performed the self-collection of NELF samples after being discharged and mailed the samples to the laboratory. (**B**) The number of asymptomatic and symptomatic paediatric and adult subjects, their severity score (0: asymptomatic; 1: mild; 2: moderate; 3: severe; 4: critically ill), age, gender, and the number of CF, NELF and plasma samples collected are shown.

**Figure 2 pathogens-11-00397-f002:**
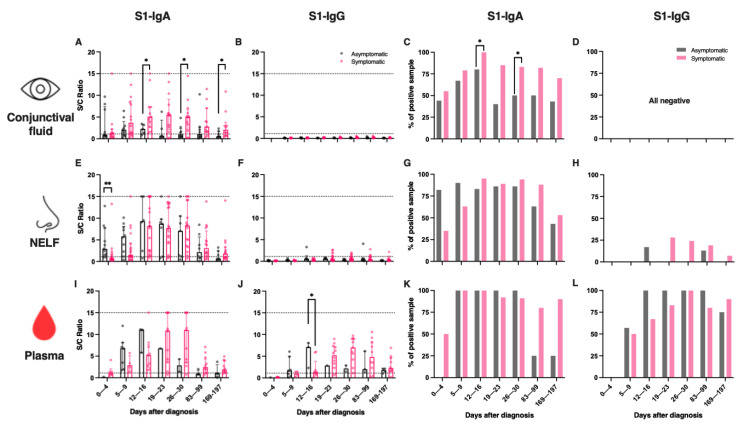
SARS-CoV-2 S1-specific antibody levels and the percentage of positive samples in the (**A**–**D**) conjunctival fluid, (**E**–**H**) nasal epithelial lining fluid (NELF), and (**I**–**L**) plasma of in asymptomatic and symptomatic paediatric patients. Grey and pink symbols indicate data from asymptomatic and symptomatic patients, respectively. Antibody-level data points above the dotted line (sample/calibrator (S/C) ratio ≥ 1.1) are considered as positive, while the dotted lines at y = 15 indicate the upper detection limit of the assay. The median and interquartile ranges are plotted, with dots representing individual values. The percentages denote the IgA and IgG positivity at each time point. The levels of S1-specific Ig were compared between asymptomatic and symptomatic patients via the Mann–Whitney test, while the percentage of positive samples was established by Fisher’s exact test at each time point. The asterisks indicate the statistical differences found; *: *p* < 0.05 and **: *p* < 0.01. “No data” indicates that the corresponding time point had no samples available.

**Figure 3 pathogens-11-00397-f003:**
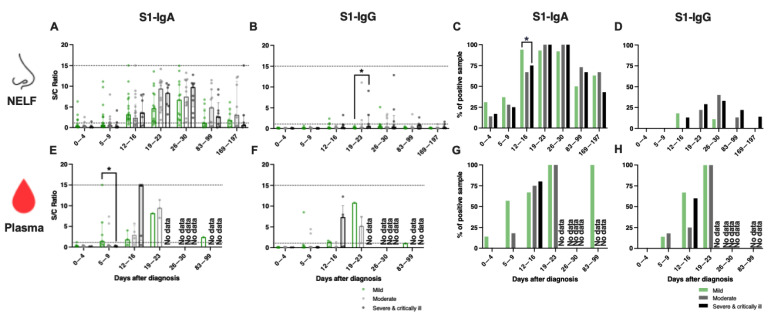
SARS-CoV-2 S1-specific antibody levels in the (**A**–**D**) nasal epithelial lining fluid (NELF) and (**E**–**H**) plasma in adult COVID-19 patients of different disease severity groups, from acute infection to the convalescent phase. Green, grey and black symbols indicate data of mild, moderate, and severe and critically ill patients, respectively. Antibody-level data points above the dotted line (sample/calibrator (S/C) ratio ≥ 1.1) are considered as positive, while the dotted lines at y = 15 indicate the upper detection limit of the assay. Median and interquartile ranges are plotted, with dots representing individual values. The percentages denote IgA- and IgG-positivity at each time point. The levels of S1-specific Ig were compared among disease severity groups with the Kruskal–Wallis test, followed by Dunn’s multiple comparisons test, with the percentage of positive samples by Fisher’s exact test at each time point. The asterisks indicate the statistical differences found, *: *p* < 0.05. “No data” labels the corresponding time point as having no samples available.

**Figure 4 pathogens-11-00397-f004:**
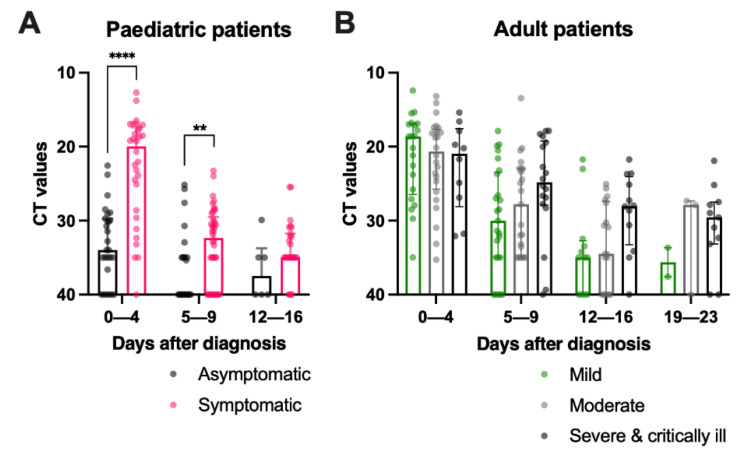
Comparison of SARS-CoV-2 viral load in (**A**) paediatric and (**B**) adult COVID-19 paediatric patients of different disease severities during hospitalization. The cycle threshold (CT) values of the SARS-CoV-2 viral gene in (**A**) asymptomatic and symptomatic paediatric patients were compared with the Mann–Whitney test and among (**B**) mild, moderate, and severe and critically ill adult patients by the Kruskal–Wallis test, followed by Dunn’s multiple comparisons test at each time point. Median and interquartile ranges are plotted with dots representing individual values. The asterisks indicate the statistical differences found, **: *p* < 0.01 and ****: *p* < 0.0001.

**Figure 5 pathogens-11-00397-f005:**
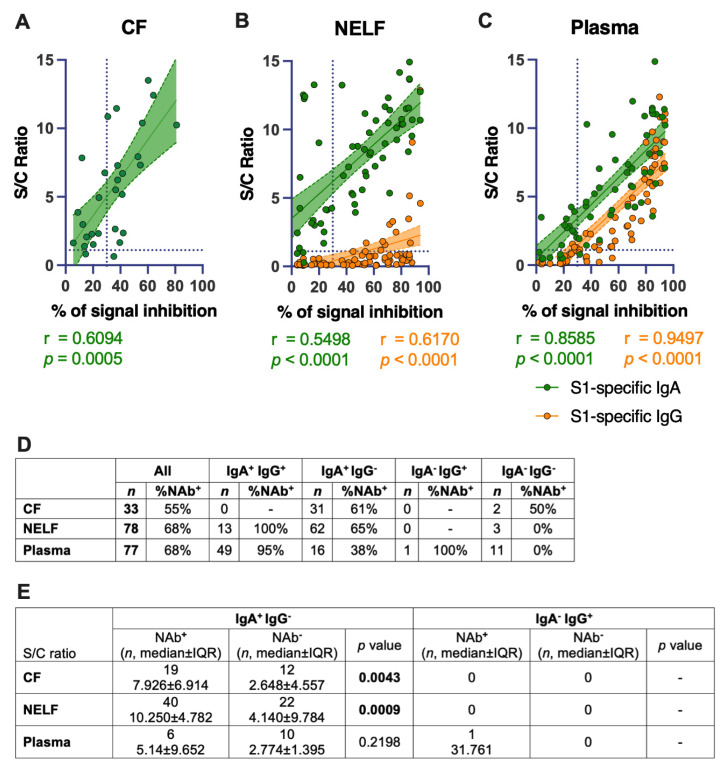
Correlation of SARS-CoV-2 S1-specific Igs to the percentage of signal inhibition in the surrogate ACE-2-based neutralization readout. (**A**) The correlation coefficients of the conjunctival fluid, (**B**) NELF, and (**C**) plasma of COVID-19 patients are superimposed on the panel, with trend lines estimated with the use of simple linear regression. Plots show the S/C ratio of the IgA (green) and IgG (orange), plotted against the percentage of inhibition of the SARS-CoV-2 spike-ACE-2 binding signal, in which an inhibition of ≥ 30% is regarded as the threshold for a positive sample, indicated by the vertical dotted line. Green and orange dotted lines represent significant linear regression fits, with 95% confidence intervals (a shaded region with the corresponding colors). (**D**) The table shows the number of each sample type included (n) in the surrogate neutralization test and the overall percentage of the sample with a neutralization effect. The number of samples with specific immunological status (e.g., IgA^+^IgG^+^) and the corresponding percentage of that immunological status with a neutralizing effect (i.e., ≥30% inhibition) are shown. (**E**) Comparison of the IgA levels between neutralizing (NAb^+^) and non-neutralizing (Nab^−^) samples with the immunological status of IgA^+^IgG^−^ was performed with a Mann–Whitney test.

**Figure 6 pathogens-11-00397-f006:**
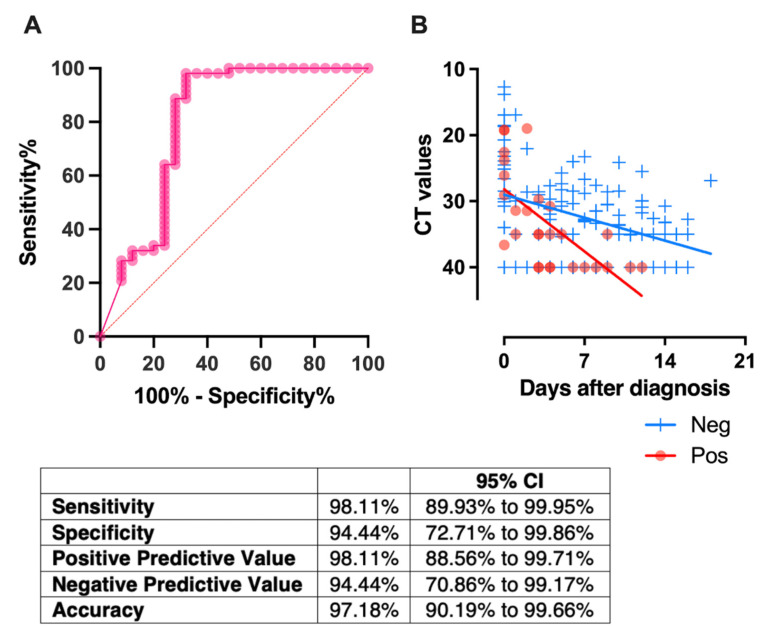
(**A**) Receiver-operating characteristic (ROC) curves, constructed using 78 NELF samples, with both IgA and NAb levels measured. The area under the curve (AUC) was 0.80 with *p* < 0.001. Using the ROC curve, the threshold for NELF IgA was defined as > 4.386 by the Youden index calculation. Using this cutoff value, the sensitivity and specificity were 98.11% and 94.44%, respectively, with an accuracy of 97.18%. (**B**) All the available CT values of the paediatric patients who showed NELF IgA levels above 
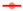
 and below 
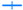
 the thresholds level are plotted against time. For this panel data, a fixed-effect regression model was applied to compare the changes of the CT values across time between these 2 groups. A statistically significant difference was found in the decline rate of the viral load at *p* = 0.002.

**Table 1 pathogens-11-00397-t001:** Comparisons of the SARS-CoV-2 S1-specific antibody levels in nasal epithelial lining fluid (NELF) between symptomatic paediatric and adult patients with mild disease.

Nasal Epithelial Lining Fluid (NELF)								
Days Post Diagnosis	IgA	0–4	5–9	12–16	19–23	26–30	83–99	169–197
Symptomatic paediatric patients	*n*	17	19	19	18	17	16	15
	Median ± IQR	0.7 ± 0.13	1.5 ± 3.1	8.2 ± 12.7	7.7 ± 8.8	8.3 ± 10.7	3.1 ± 5.6	1.8 ± 2.4
	%	35	63	95	89	94	88	53
Adult patients with mild disease	*n*	13	17	18	15	14	10	8
	Median ± IQR	0.3 ± 1.7	1.1 ± 4.0	3.4 ± 5.5	4.7 ± 6.7	5.9 ± 8.5	1.1 ± 4.2	1.7 ± 3.9
	%	31	47	89	93	93	50	63
Mann-Whitney test between groups		0.3203	0.2676	0.1701	0.1004	0.2619	0.1089	0.8746
Fisher’s Exact test between groups		ns	0.5027	0.6039	ns	ns	0.0687	ns
	**IgG**							
Symptomatic paediatric patients	*n*	16	18	18	18	17	16	15
	Median ± IQR	0.1 ± 0.0	0.1 ± 0.1	0.2 ± 0.2	0.5 ± 1.2	0.6 ± 1.0	0.4 ± 0.8	0.2 ± 0.3
	%	0	0	0	28	24	19	7
Adult with mild disease	*n*	10	10	10	10	9	8	7
	Median ± IQR	0.1 ± 0.1	0.1 ± 0.1	0.1 ± 1.0	0.2 ± 0.2	0.6 ± 0.8	0.1 ± 0.3	0.1 ± 0.1
	%	0	0	20	0	11	0	0
Mann-Whitney test between groups		0.8664	0.1632	0.2400	0.0821	0.6723	**0.0034**	**0.0085**
Fisher’s Exact test between groups		ns	ns	0.1190	0.1282	0.6279	0.5257	ns

**Table 2 pathogens-11-00397-t002:** Comparisons of the SARS-CoV-2 S1-specific antibody levels in plasma between symptomatic paediatric and adult patients with mild disease.

Plasma								
Days Post Diagnosis	IgA	0–4	5–9	12–16	19–23	26–30	83–99	169–197
Symptomatic paediatric patients	*n*	6	4	9	12	11	10	10
	Median ± IQR	1.1 ± 1.4	2.9 ± 3.6	5.2 ± 5.3	10.9 ± 11.2	11.0 ± 11.5	2.5 ± 3.7	1.9 ± 2.8
	%	50	100	100	92	91	80	90
Adult patients with mild disease	*n*	7	8	4	1	0	1	0
	Median ± IQR	0.2 ± 0.3	2.2 ± 4.9	2.9 ± 7.3	8.2		2.4	-
	%	14	63	75	100	-	100	-
Mann-Whitney test between groups	*p* value	**0.0082**	0.8081	0.5035	-	-	-	-
Fisher’s Exact test between groups	*p* value	0.2657	0.4909	0.3077	-	-	-	-
	**IgG**							
Symptomatic paediatric patients	*n*	6	4	9	12	11	10	10
	Median ± IQR	0.1 ± 0.1	1.0 ± 1.1	1.3 ± 3.0	5.2 ± 4.7	7.0 ± 4.8	4.8 ± 6.6	2.3 ± 3.2
	%	0	50	67	83	100	80	90
Adult with mild disease	*n*	7	8	4	1	0	1	0
	Median ± IQR	0.1 ± 0.0	0.3 ± 0.7	1.5 ± 1.8	10.8	-	1.1	-
	%	0	13	75	100	-	0	-
Mann-Whitney test between groups	*p* value	0.0734	0.2828	0.8252	-	-	-	-
Fisher’s Exact test between groups	*p* value	ns	0.2364	ns	-	-	-	-

**Table 3 pathogens-11-00397-t003:** Comparisons of the CT values of SARS-CoV-2 among patient groups, along with time points.

A. CT Values						
Days Post-Diagnosis		0–4	5–9	12–16	19–23	26–30
Paediatric patients						
Asymptomatic	**n**	29	23	6	0	0
	Median range	34.0 22.6–40.0	40.0 25.12–40.0	37.5 29.9–40.0	-	-
Symptomatic	**n**	30	42	24	1	0
	Median range	20.0 12.7–27.3	32.3 23.3–40.0	35.0 25.4–40.0	40.0 -	-
Mann–Whitney test between groups	*p* value	**<0.0001**	**0.0234**	0.2218	-	-
Adult patients						
Mild disease	**n**	21	25	12	2	0
	Median range	18.6 12.4–35.0	30.0 17.9–40.0	35.0 21.7–40.0	35.6 33.7–37.6	-
Moderate disease	**n**	24	23	17	3	1
	Median range	20.7 13.2–35.3	27.8 13.4–35.0	34.5 25.1–40.0	27.9 27.4–40.0	35.0 -
Severe and critically ill	**n**	10	18	12	11	1
	Median range	21.0 15.4–32.1	24.8 17.9–40.0	28.0 21.7–40.0	29.6 21.9–40.0	30.0 -
Kruskal–Wallis test among adult groups	*p* value	0.7533	0.0938	0.0689	0.4966	-
Kruskal–Wallis test among all groups	*p* value	**<0.0001**	**<0.0001**	**0.0157**	-	-
Dunn’s multiple comparisons test						
*Asymptomatic vs symptomatic paediatric patients*		**<0.0001**	0.6680	ns	-	-
*Asymptomatic paediatric patients* vs. *adult patients*—*mild disease*		**<0.0001**	**0.0107**	ns	-	-
*Asymptomatic paediatric patients* vs. *adult patients*—*moderate disease*		**<0.0001**	**0.0004**	ns	-	-
*Asymptomatic paediatric patients* vs. *adult patients*—*severe and critically ill*		**0.0024**	**<0.0001**	**0.0328**	-	-
*Symptomatic paediatric patients* vs. *adult patients*—*mild disease*		ns	0.631	ns	-	-
*Symptomatic paediatric patients* vs. *adult patients*—*moderate disease*		ns	**0.0466**	ns	-	-
*Symptomatic paediatric patients* vs. *adult patients*—*severe and critically ill*		ns	**0.0007**	**0.0361**	-	-
*Adult patients: mild disease* vs. *moderate disease*		ns	ns	ns	-	-
*Adult patients: mild disease* vs. *severe and critically ill*		ns	0.3646	ns	-	-
*Adult patients: mild disease* vs. *severe and critically ill*		ns	ns	ns	-	-

The CT values of the SARS-CoV-2 viral gene were expressed in median and the data range. More than one specimen might have been collected from the same patients within the same period, therefore, the sample number analysed at each time point might exceed the number of research subject within the patient group. A detail breakdown of the sample size collected on each day post diagnosis is available in Appendix A. The comparisons between the asymptomatic and symptomatic paediatric subjects were tested by Mann-Whitney test while those among the three adult patients’ groups and among all groups were done by Kruskal-Wallis test followed by Dunn’s multiple comparisons test.

**Table 4 pathogens-11-00397-t004:** Comparisons of the Ct values of SARS-CoV-2 of the same patient groups between time points.

Days Post-Diagnosis	0–4 vs. 5–9	0–4 vs. 12–16	0–4 vs. 19–23	5–9 vs. 12–16	5–9 vs. 19–23	12–16 vs. 19–23
Asymptomatic paediatric patients	0.0564	ns	-	ns	-	-
Symptomatic paediatric patients	ns	**<0.0001**	**-**	**<0.0001**	-	-
Adult patients—mild disease	**<0.0001**	**0.0006**	-	ns	-	-
Adult patients—moderate disease	**0.0012**	**0.0053**	**0.0075**	ns	ns	ns
Adult patients—severe & critically ill	ns	ns	0.0593	ns	0.1322	0.8198

The comparisons between the time points of the same patient groups were compared with the Kruskal–Wallis test, followed by Dunn’s multiple comparisons test. *p* values smaller than 0.05 are bolded, *p* values > 0.9999 are represented by ns (not significant).

## Data Availability

The datasets generated are available from the corresponding author on reasonable request.

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
