# Peer review of "Mucosal Antibody Response to SARS-CoV-2 in Paediatric and Adult Patients: A Longitudinal Study"

_pathogens, 2022, doi:10.3390/pathogens11040397_

Round 1
Reviewer 1 Report
This paper describes the kinetics of the ocular and nasal mucosal-specific immunoglobulin A (IgA) produced in response to SARS-CoV-2 infection. It is an interesting topic that has never been explored in full and is very relevant concerning the protective role of the immune system at the mucosal level whether induced by natural infection or vaccination. The manuscript has several weaknesses, e.g., no titration of neutralizing antibodies, no information concerning past infections by other coronaviruses, and others cited by the authors, but the results are interesting.
The Authors shall address the following:
- Provide more information about the RT-PCR tests used. Are the kits IVD approved? What is the maximum number of amplification cycles performed by the tests? Samples scoring positive at 45 or 47 cycles are nonsense as RT-PCR after 40 cycles is not reliable anymore.
- I guess none of these patients were vaccinated but it is not stated. Please describe the vaccine status of the examined subjects.
- What were the comorbidities in pediatric and elderly patients suffering severe infection?
- Please compare your results with those of vaccinated people if available in the literature. If not, it would have been nice to collect a few samples from vaccinated and non-infected (either currently or previously) and compare IgA and IgG titers in the mucosal fluids.
Author Response
Please see the attachment, thanks.

Reviewer 2 Report
Very interesting work, the authors did a great job to characterize the mucosal antibody response against SARS2.
Minor comments:
- line 171, should be 3B and 3D
- did any of the adult patients receive vaccination? and any of patients were infected before diagnosed as positive again in this current study?
Author Response
Q1: line 171, should be 3B and 3D
A1: Thanks for the reminder, the typos have been amended.
Q2: did any of the adult patients receive vaccination? and any of patients were infected before diagnosed as positive again in this current study?
A2: No vaccinated subjects and no repeated infection, these have been clarified in the revised methodology.
Reviewer 3 Report
The authors assessed the IgA and IgG antibody levels in the two patient groups, paediatric patients and adult patients. Three sample types were assessed, (1) conjunctival fluid, (2) nasal epithelial lining and (3) plasma. Variables (e.g. patient groups, disease severity, days after symptom onset, neutralizing ability) were assessed to find out if there were correlation with the antibody levels. The research gaps were well defined and the limitations of the study were addressed. The study will be useful for the colleagues in the field, however, the organization of the manuscript should be improved further. Otherwise, readers will be difficult to follow the messages conveyed by the authors.
Specific comments:
- Duplicate information.
- redundant information presented in Table 1 vs Figure 2 (they both presented IgA and IgG levels in paediatric patients)
- redundant information presented in Table 2 vs Figure 3 (they both presented IgA and IgG levels in adult patients).
- the authors can keep the figures only and put the tables in supplementary section.
- Figure 2 and Figure 3 can be combined together for easy comparison between paediatric patients and adult patients respectively.
- Organize the data again. Describe the antibody levels in each patient group first:
Antibody levels in paediatric patients
(i) IgA in (1) conjunctival fluid, (2) nasal epithelial lining and (3) plasma
(ii) IgG in (1) conjunctival fluid, (2) nasal epithelial lining and (3) plasma
Antibody levels in adult patients
(i) IgA in (1) nasal epithelial lining and (2) plasma
(ii) IgG in (1) nasal epithelial lining and (2) plasma
Then compare and contrast the two different patient groups.
- Compare the neutralizing ability of IgA and IgG using the GenScript kit.
Among IgA positive samples, % of samples with neutralizing ability.
(i) IgA in (1) conjunctival fluid, (2) nasal epithelial lining and (3) plasma
(ii) IgG in (1) conjunctival fluid, (2) nasal epithelial lining and (3) plasma
Antibody levels in adult patients
(i) IgA in (1) nasal epithelial lining and (2) plasma
(ii) IgG in (1) nasal epithelial lining and (2) plasma
Then compare and contrast the two different patient groups.
- Delete section 2.8 and Table 4, they are irrelevant to your study.
- The number of samples collected and analyzed were not concordant.
For example, CF samples from pediatrics patients were not matched.
- line 91 and Figure 1B: 140 CF samples collected
- Table 1A: 133 CF samples were analyzed (20+22+17+18+20+19+17)
The authors have to check for other samples when necessary and explain the underlying reasons.
- Table 1 should be revised for clarity.
- suggest to add a column ‘All’
- suggest to add a table to show all zero results for CF-IgG (to be concordant with the figure presented)
- Method section. After going through the ‘Instructions for Authors’ from the ‘Pathogens’, there is no word limit. It is not necessary to separate the methods into two, just combine the supplementary method into the main text. It is not convenient for readers to go back and forth to realize the study design. Then, the manuscript will look neat and tidy. Beware of the subsection title do not match with the contents. For example, under ‘Subject recruitment and Severity Scoring’ in supplementary method, you mentioned specimens collection. They should not be into this section. Need to write down the study period for subject recruitment (e.g. XX month 2020 to YY month 2021).
Minor comments:
- It is clumsy to mention S1-specific throughout the manuscript. It is ok that you mention IgA and IgG in condition that you documented the methods to asses them in the method section.
- the term used in section 2.9 ‘SARS-CoV-2 receptor-blocking effect’ can be revised to ‘neutralizing ability’, readers will know in the method section that the GenScript kit was used as a surrogate marker for assessing neutralizing ability.
Author Response
Please see the attachment, thanks

Round 2
Reviewer 3 Report
The authors addressed all of my queries.